# Anticancer Effect of Gallic Acid on Acidity-Induced Invasion of MCF7 Breast Cancer Cells

**DOI:** 10.3390/nu15163596

**Published:** 2023-08-16

**Authors:** Ran Hong, Sung-Chul Lim, Tae-Bum Lee, Song-Iy Han

**Affiliations:** 1Department of Pathology, College of Medicine, Chosun University, Gwangju 61452, Republic of Korea; nanih@chosun.ac.kr (R.H.); sclim@chosun.ac.kr (S.-C.L.); 2Division of Premedical Science, College of Medicine, Chosun University, Gwangju 61452, Republic of Korea; tblee01@chosun.ac.kr

**Keywords:** gallic acid, MCF7 cells, acidic tumor environment

## Abstract

The acidic tumor environment has emerged as a crucial factor influencing the metastatic potential of cancer. We investigated the effect of an acidic environment on the acquisition of metastatic properties in MCF7 breast cancer cells and explored the inhibitory effects of gallic acid. Prolonged exposure to acidic culture conditions (over 12 weeks at pH 6.4) induced the acquisition of migratory and invasive properties in MCF7 cells, accompanied by increased expression of Matrix Metalloproteinase 2 and 9 (MMP2 and MMP9, respectively), together with alterations in E-cadherin, vimentin, and epithelial-to-mesenchymal transition markers. Gallic acid effectively inhibited the survival of acidity-adapted MCF7 (MCF7-6.4/12w) cells at high concentrations (>30 μM) and reduced metastatic characteristics induced by acidic conditions at low concentration ranges (5–20 μM). Moreover, gallic acid suppressed the PI3K/Akt pathway and the nuclear accumulation of β-catenin, which were elevated in MCF7-6.4/12w cells. These findings highlight the potential of gallic acid as a promising therapeutic agent for metastatic traits in breast cancer cells under acidic conditions.

## 1. Introduction

Breast cancer is one of the most prevalent cancers worldwide and is a leading cause of cancer-related deaths among women [1]. Patients with breast cancer have a high risk of metastasis throughout their lives because of the strong metastatic nature of breast cancer cells [2]. During metastatic progression, tumor cells acquire new features such as migration and invasion [3]. Despite significant advancements in diagnosis and treatment, the metastatic potential of breast cancer remains a challenge.

In particular, the tumor microenvironment has emerged as a critical factor that influences the metastatic potential of cancer [4]. A notable characteristic of the tumor microenvironment is acidification, which often results in the development of an acidic milieu, known as acidosis. Acidosis can occur because of the accumulation of acid byproducts resulting from vigorous energy generation and altered glucose metabolism in tumor cells [5]. Inadequate blood supply and dysfunctional pH regulation also contribute to this phenomenon. Recent studies have suggested that acidic microenvironments are associated with increased invasiveness, angiogenesis, and therapy resistance in various cancer types including breast cancer, highlighting acidosis as a potential target for therapeutic intervention [6,7]. 

Considering the limitations and side effects associated with conventional cancer treatments, the exploration of natural compounds as alternative therapeutic agents has gained considerable attention. Natural products have long been recognized as valuable sources of bioactive compounds with anticancer properties. Identifying natural compounds capable of mitigating the invasive behavior and reducing the viability of breast cancer cells under acidic conditions could provide novel possibilities for therapeutic development. 

Gallic acid (3,4,5-trihydroxybenzoic acid) is a polyphenolic compound that is commonly found in fruits, vegetables, and medicinal plants. It acts as an antioxidant and exhibits antiviral, antibacterial, and antifungal properties, making it a widely used natural remedy [8]. More importantly, numerous studies have suggested its beneficial effects in inhibiting tumor growth and metastasis in various cancer types [9,10]. Gallic acid displays diverse effects at different molecular levels in various tumor types, highlighting its potential as an important bioactive drug for therapeutic applications. However, the specific effects of gallic acid on breast cancer cells cultured in acidic environments have not yet been investigated. 

Among the crucial signaling pathways that play a significant role in regulating cancer growth and metastasis, dysregulation of the PI3K/Akt pathway is commonly observed in various types of cancers, including breast cancer. Although the PI3K/Akt pathway is indispensable for controlling normal cellular processes in response to growth factors and signaling molecules, overactivation of this pathway is linked to increased tumor malignancy and poor prognosis [11]. On the other hand, β-catenin is a crucial regulator during normal embryonic development; however, in cancer cells, it has a pivotal role, driving cancer stemness, epithelial-to-mesenchymal transition (EMT), and metastasis. In normal cellular conditions, β-catenin is expressed at low levels and located at the cell membrane, regulating cell adhesion. However, under abnormal signaling, protein levels increase, causing β-catenin to translocate into the cell nucleus, where it promotes the expression of essential genes required for cancer metastasis [12].

In this study, we discovered that nonmetastatic MCF7 breast cancer cells can develop metastatic properties when subjected to extended periods of acidic culture conditions. Furthermore, we investigated the potential of gallic acid to attenuate these metastatic traits by inhibiting the elevated PI3K/Akt pathway and nuclear accumulation of β-catenin.

## 2. Materials and Methods

### 2.1. Cell Lines and Culture Conditions

MCF7 human breast cancer cells were obtained from the Korean Cell Line Bank (Seoul, Korea) and cultured in RPMI 1640 medium (Invitrogen, Carlsbad, CA, USA) supplemented with 10% (*v*/*v*) fetal bovine serum and 1% penicillin–streptomycin at 37 °C in a 5% CO_2_ atmosphere. All chemicals were obtained from Calbiochem (San Diego, CA, USA). Acidic media were prepared by adding 5 N HCl to adjust the pH to 6.0, 6.2, 6.4, and 6.6, which were then followed by filter sterilization immediately before usage.

### 2.2. Invasion and Migration Assay

Matrigel-coated Transwell chambers (Corning Costar) were used for the cell invasion assay. Equal numbers of cells were suspended in a medium containing 1% FBS. Approximately 200 μL of the cell suspension was added to the upper part of the insert, and medium containing 5% FBS was added to the lower portion of the insert. After 24 h of incubation at 37 °C in a 5% CO_2_ atmosphere, noninvasive cells on the upper surface of the Transwell membrane were removed with a cotton swab, and the invaded cells on the lower surface were fixed with 4% formaldehyde and stained with crystal violet solution. The number of invading cells was counted or imaged using a high-power microscope at 200× magnification (Olympus corporation, Tokyo, Japan). For the migration assay, cells incubated at pH 7.4 and 6.5 were treated as mentioned above and grown in 24-well plates in growth medium. After an overnight culture, a wound of constant width was created in the middle of the cell surface by scraping with a micropipette tip. The debris was washed with PBS, and wound closure was monitored and photographed at 24 h under a microscope at 100× magnification (Olympus corporation, Tokyo, Japan). The migration distance was analyzed using HMIAS-2000 software version 2.0 (Qianping Image Technology, Wuhan, China).

### 2.3. Real-Time Reverse Transcription–Polymerase Chain Reaction

Real-time PCR was conducted using the Light Cycler 2.0 system (Roche) with the Fast Start DNA Master SYBR Green I Kit (Roche) to amplify the target genes. To verify the correct amplification products, the PCR products were analyzed on 2% agarose gel stained with ethidium bromide. The primer sequences for β-actin, E-cadherin, vimentin, and MMP2 were as follows: 5′-GACTATGACTTAGTTGCGTTA-3′ and 5′-GCCTTCATACATCTCAAGTTG-3′, 5′-TTGAGGCCAAGCAGCAGTA-3′ and 5′-CGCCAAAGTCCTCGGACA-3′, 5′-TTCAGAGAGAGGAAGCCGA-3′ and 5′-ATGCTGTT CTGAATCTGAG-3′, and 5′-CGACCGCGACAAGAAGTA-3′, and 5′-GCACACCACATCTTTCCGTCA-3′. Primers for MMP9 (P323207) were purchased from Bioneer (Daejeon, Korea). PCR was performed with initial denaturation at 95 °C for 10 min, followed by 45 cycles of denaturation at 95 °C for 15 s, annealing at 60 °C for 5 s, and extension at 72 °C for 7 s. Melt curve analysis was conducted to confirm the presence of a single amplification product. Negative controls without template were included in each run. Data analysis was performed using Light Cycler software (version 4.0; Roche, Switzerland), and the 2^–ΔΔCt^ method was utilized to analyze relative gene expression.

### 2.4. Western Blot Analysis

Cells treated under specific conditions were lysed using a whole-cell lysis buffer (50 mM HEPES, 150 mM NaCl, 1% Triton X-100, 5 mM EGTA, and a protease inhibitor cocktail). Equal amounts of protein extracts were separated by 10% sodium dodecyl sulfate-polyacrylamide gel electrophoresis (SDS-PAGE) and transferred to nitrocellulose membranes using standard techniques. Membranes were probed with specific antibodies against MMP2 (ab86607, Abcam, Cambridge, UK), MMP9 (sc-376861, Santa Cruz Biotechnology, Santa Cruz, CA, USA), E-cadherin (sc-8426), vimentin (sc-6260), β-catenin (sc-7963), p85 (#sc-1637, Cell Signaling Technology, Danvers, MA, USA) Akt (#4691), p-Akt (#4060), and α-tubulin (sc-5286). The antibodies were diluted in TBS solution containing 2% skim milk and incubated overnight at 4 °C. Signals were detected using an Image Station 4000MM image analyzer (Kodak, Rochester, NY, USA).

### 2.5. Cytotoxicity Assay

The EZ-Cytox viability assay was performed according to the manufacturer’s instructions. Cells were seeded in 24-well plates at a density of 5–8 × 10^4^ cells/well and allowed to grow for 24 h. Afterward, the cells were exposed to either the growth medium with or without gallic acid for a 48 h incubation period. Following this, the EZ-Cytox solution (Daeillab, Korea) was added to the wells and incubated at 37 °C in a CO_2_ incubator for the final 2 h of incubation. The absorbance of the samples was measured at 450 nm using an enzyme-linked immunosorbent assay (ELISA) plate reader. The absorbance of untreated cells was set to 100%, and cell survival was expressed as a percentage relative to this value.

### 2.6. Apoptosis Analysis

The cells were treated with gallic acid, and during the final 10 min of treatment, they were stained with Hoechst 33342 at a concentration of 1 µg/mL. After treatment, both floating and attached cells were collected by centrifugation, and the resulting cell pellets were washed with ice-cold PBS. Subsequently, the cells were fixed on ice using 3.7% formaldehyde solution for 15 min and washed again with PBS. To prepare slides, a fraction of the cell suspension was centrifuged using a Shandon centrifuge (Thermo Fisher Scientific, Waltham, MA, USA). The resulting slides were air-dried and mounted in an antifade solution. The slides were then examined under a fluorescence microscope (DM5000, Leica, Wetzlar, Germany) to identify apoptotic nuclei. A total of 500 cells, randomly selected from different fields, were counted, and the percentage of apoptotic cells was calculated by dividing the number of apoptotic cells by the total number of cells.

### 2.7. Statistical Analysis

Numerical data are expressed as the mean ± standard error (SE) of three independent experiments. Student’s *t* test was used for simple comparisons, and one-way analysis of variance (ANOVA) with Tukey’s test was used for multiple comparisons. A *p*-value < 0.05 was considered statistically significant.

## 3. Results

### 3.1. Induction of Metastatic Properties in MCF7 Cells by Long-Term Exposure to Environmental Acidity

For this study, we used MCF7, a breast adenocarcinoma cell line with a differentiated mammary epithelium and few metastatic characteristics [13]. To investigate the impact of acidic pH conditions on the invasive ability of MCF7 cells, the cells were exposed to normal pH medium (pH 7.4) and various pH-adjusted acidic culture media (pH 6.6, 6.4, 6.2, and 6.0) for up to 3 d. Invasive activity was analyzed using a Matrigel-coated Transwell plate. However, compared to cells cultured at pH 7.4, no significant increase was observed in the number of cells that invaded the lower surface of the Transwell membrane in MCF7 cells exposed to various acidic culture conditions (Figure 1A).

Next, we investigated whether prolonged exposure to acidic culture conditions induces the invasive ability of MCF7 cells. The cells were maintained for several weeks under acidic pH conditions of 6.6, 6.4, 6.2, and 6.0, and subculturing was performed every 4–5 d. As the duration of incubation in acidic culture medium increased, the cell viability decreased significantly, and cells cultured under acidic conditions at pH 6.0 and 6.2 exhibited cell death upon extended incubation. Thus, a long-term study was conducted at pH 6.4, which is the lowest pH at which cell viability could be maintained. Initially, the proliferation of cells cultured at pH 6.4 for the initial 4 weeks was found to progress very slowly. However, it gradually recovered thereafter, and after 12 weeks of culture, the cell growth rate approached approximately 80% of that observed in the cells cultured under normal pH conditions (Figure 1B). With an extended culture period in an acidic medium at pH 6.4, the cellular morphology underwent a significant transformation, adopting an elongated filopodia-like shape reminiscent of fibroblasts (Figure 1C). Furthermore, MCF7 cells cultured at pH 6.4 condition for 12 weeks displayed migratory and invasive properties (Figure 1D–G).

To investigate the molecular mechanisms underlying the acquisition of motility and invasive traits in acidic environments, we performed experiments using MCF7 cells cultured for 12 weeks at pH 6.4 (MCF7-6.4/12w).

In this study, we examined the expression of Matrix Metalloproteinases (MMPs), which play crucial roles in the migration and invasion of cancer cells by degrading extracellular matrix proteins. Using real-time PCR and immunoblotting, we observed a significant increase in the expression of MMP2 and MMP9 in cells subjected to prolonged acidic culture (Figure 2A,B,E). Furthermore, we examined the expression of E-cadherin and vimentin, known EMT markers. We observed a decrease in E-cadherin expression and an increase in vimentin expression in MCF7-6.4/12w cells (Figure 2C–E). These findings suggest that the migratory and invasive properties of MCF7-6.4/12w cells may be associated with MMPs expression and an epithelial-to-mesenchymal transition EMT-like mechanism.

### 3.2. Inhibitory Effect of Gallic Acid on Survival of Both Normal and Acidity-Adapted MCF7 Cells 

As shown in Figure 1C, tumor cells adapted to acidic culture conditions exhibited restored growth rates. Such adaptations to acidic environments often lead to the development of resistance, making treatment challenging. Previous studies have shown that cancer cells exposed to acidic media display strong resistance to several commonly used anticancer drugs, including doxorubicin, daunorubicin, cisplatin, oxaliplatin, etoposide, and 5-FU [14]. Therefore, the current study aimed to identify substances capable of effectively controlling tumor cells that have adapted to acidic conditions. Through a comprehensive evaluation of natural polyphenol compounds known for their anticancer activities, we found that gallic acid inhibited the survival of MCF7-6.4/12w cells. Initially, we assessed the viability of normal MCF7 cells and MCF7-6.4/12w cells using EZ-Cytox viability assay across a concentration range of 0–100 μM. Treatment with increasing concentrations of gallic acid >30 μM significantly reduced the survival of both normal MCF7 cells and MCF7-6.4/12w cells in a dose-dependent manner (Figure 3A). Moreover, gallic acid treatment at concentrations >50 μM significantly increased apoptotic body formation in both MCF7 and MCF7-6.4/12w cells (Figure 3B–D). These results indicate that gallic acid can effectively suppress the growth and viability of not only normal MCF7 cells but also acidity-adapted MCF7-6.4/12w cells. 

### 3.3. Low Concentrations of Gallic Acid Decreases Acidity-Induced Metastatic Characteristics in MCF7-6.4/12w Cells

Numerous studies have elucidated the inhibitory properties of gallic acid against metastatic traits, including cell migration and invasion of cancer cells [15,16]. Based on these findings, our study aimed to explore the effects of low concentrations of gallic acid (<30 μM) on the invasive ability of MCF7-6.4/12w cells, which have adapted to acidic culture environments. Notably, at low concentrations, gallic acid had no significant effect on cell viability. After treatment with gallic acid at concentrations of 5, 10, and 20 μM for 48 h, the invasion of MCF7-6.4/12w cells was significantly reduced by 37, 54, and 78%, respectively (Figure 4A,B). Consistently, treatment with low concentrations of gallic acid decreased the expression of MMP2, MMP9, and vimentin, and partially restored E-cadherin expression in MCF7-6.4/12w cells (Figure 4C–F). Thus, our findings suggest that a low dose of gallic acid effectively suppresses the metastatic characteristics of acidity-adapted MCF7-6.4/12w cells.

### 3.4. Gallic Acid Inhibits β-Catenin Nuclear Accumulation in MCF7-6.4/12w Cells

To understand the mechanisms by which gallic acid interferes with the metastatic features of MCF7-6.4/12w cells, we examined the expression and localization patterns of β-catenin in MCF7-6.4/12w cells. β-catenin is a crucial factor in cancer cell motility and metastasis, as it induces EMT and promotes the formation of cancer stem cells. Under normal conditions, β-catenin is present at low levels and in an inactive form in the cytosol; upon activation, it translocates to the nucleus, triggering transcriptional activity. Under these conditions, MCF7-6.4/12w cells exhibited upregulation of β-catenin protein levels accompanied by its localization in the nucleus (Figure 5A). Notably, treatment with gallic acid decreased this effect, leading to a reduction in total and nuclear β-catenin accumulation (Figure 5B). 

### 3.5. Gallic Acid Inhibits the PI3K/Akt Pathway Involved in MCF7-6.4/12w Cell Invasion

The PI3K/Akt pathway plays a vital role in the survival, proliferation, invasion, and overall progression of cancer cells. Previous studies have indicated that cell motility can be enhanced through Akt activation under acidic conditions [7,17], and that gallic acid can inhibit cancer cell growth and malignancy by controlling Akt in various cancer types, such as bladder cancer and myeloid leukemia [9,18,19,20]. 

In this study, we evaluated whether the PI3K/Akt pathway is associated with the anti-invasive effects of gallic acid in acidity-adapted MCF7-6.4/12w cells. When we measured the protein levels of p85, a regulatory subunit of PI3K, in MCF7 and MCF7-6.4/12w cells, higher levels of p85 were detected in MCF7-6.4/12w cells than MCF7 cells (Figure 5C). Consistently, Akt, a downstream effector of PI3K, was highly phosphorylated in MCF7-6.4/12w cells compared with normal MCF7 cells (Figure 5C). The elevated levels of p85 and p-Akt were partially suppressed by 10 or 20 μM of gallic acid treatment (Figure 5D). Additionally, the use of LY294002 or AKT inhibitor VIII to inhibit the PI3K/Akt pathway led to a significant reduction in invasion (Figure 5E,F) and in total and nuclear β-catenin levels in MCF7-6.4/12w cells (Figure 5G).

These results indicate that the metastatic properties of MCF7-6.4/12w cells are associated, at least partially, with the activation of β-catenin, which is regulated by the PI3K/Akt pathway, and can be effectively suppressed by gallic acid. Figure 6 presents an overview of the mechanism by which gallic acid counteracts metastatic traits induced by prolonged exposure to acidity.

## 4. Discussion

Acidosis, a metabolic parameter in tumor tissue, has emerged as a significant driver of invasion and metastasis in various cancer cell types, including melanoma, glioma, and prostate cancer [21,22,23]. Certain metastatic cancer cells exhibit increased migration and invasive potential upon exposure to acidic conditions for short periods [24]. Our previous studies also demonstrated that metastatic gastric cancer cell lines exhibit elevated malignant phenotypes and develop resistance to chemical drugs after exposure to acidic culture condition for 48–72 h [14,25].

Building upon these findings, we investigated whether acidic conditions could induce invasion in nonmetastatic MCF7 cells. Given that MCF7 cells present features of differentiated mammary epithelium with limited metastatic potential [13], they serve as valuable models for studying the process of metastatic transition under specific circumstances [26]. Previous studies have revealed that MCF7 cells exhibit more aggressive features under specific conditions, such as hypoxia or induction of CD144 expression [27,28].

In this study, despite the initial lack of invasive characteristics, prolonged exposure to acidity led to the acquisition of invasive properties in MCF7 cells. Previously, it was shown that environmental acidity enhanced the motility of MCF7 cells co-cultured with fibroblasts in spheroids [29]. Taken together with our finding, these results suggest that even nonmetastatic MCF7 cells can adopt metastatic traits under acidic conditions. This process seems to be influenced by alterations in signaling pathway and gene expression, which occur during exposure to additional stimuli or extended acidic environments.

Since tumor acidity is intricately linked to poor prognosis of cancer, considerable efforts have been made to address the challenges posed by acidic tumor environments in cancer treatment. One approach involves combining proton pump inhibitors or systemic buffering agents to regulate the acidity within the tumor microenvironment which would enhance the effectiveness of treatments [30,31]. However, the effectiveness of these approaches was not as significant as anticipated. As an alternative approach, we endeavored to identify candidate agents that could maintain efficacy even under acidic conditions and effectively inhibit metastatic pathways. Thus, we investigated the therapeutic potential of gallic acid in breast cancer cells that have undergone phenotypic changes due to adaptation to an acidic environment. Previous studies have suggested that the anticancer activity of gallic acid is associated with the induction of cell apoptosis through various mechanisms, including the generation of reactive oxygen species, regulation of the glutathione system, and control of apoptotic and antiapoptotic proteins, and cell cycle regulators [32,33,34,35].

Our study also revealed that gallic acid can induce apoptosis in both normal MCF7 cells and acid-adapted MCF7-6.4/12w cells, at high concentrations >50 μM. However, we focused on the inhibitory effect of low concentrations of gallic acid on the metastatic properties, as potential toxicity concerns are associated with high-dose treatments. Still, even so, we can expect significant anticancer effects by simply inhibiting metastasis.

To ensure cellular viability, we carefully selected a gallic acid concentration that maintained a survival rate >90% after 48 h of incubation. Remarkably, even at this comparatively low concentration, gallic acid effectively suppressed the expression of MMP2 and MMP9 and inhibited the Matrigel invasion capability of MCF7-6.4/12w cells. Additionally, gallic acid treatment facilitated the recovery of E-cadherin expression and suppressed vimentin expression. The effects were statistically significant, despite being partial. These findings indicate that gallic acid exerts inhibitory effects on EMT. Such mechanisms appear to be associated with the suppression of increased accumulation and nuclear translocation of β-catenin, along with the downregulation of PI3K and phospho-Akt in MCF7-6.4/12w cells. The suppression of β-catenin accumulation due to inhibition of the PI3K/Akt pathway suggests that excessive PI3K/Akt pathway activation in cells adapted to an acidic environment may induce β-catenin activation. While β-catenin activity is induced by Wnt activation in the canonical pathways, it is also regulated by the Akt pathway. Moreover, gallic acid appeared to partially suppress the PI3K/Akt pathway, a significant signaling pathway that regulates the activation and expression of various cancer-related factors. Our results are in line with those of previous studies, suggesting that gallic acid exhibits anticancer activity by inhibiting the PI3K/Akt pathway in different cancer cell types, including lung cancer, bladder cancer, and myeloid leukemia [19,20,36]. In addition, gallic acid has shown tumor suppressive effects in osteosarcoma, hepatocellular carcinoma, and gastric precancerous regions by regulating β-catenin [37,38,39]. Therefore, gallic acid has the potential to exert powerful anticancer effects in various cancer types, given its ability to effectively inhibit key regulators of cancer control, including the PI3K/Akt and β-catenin pathways.

## 5. Conclusions

This study revealed the detrimental influence of prolonged exposure to an acidic environment, intensifying the aggressiveness of cancer by conferring invasiveness to nonmetastatic breast cancer cells. Furthermore, treatment with gallic acid has demonstrated a remarkable ability to effectively suppress these traits, specifically by targeting the PI3K/Akt pathway and reducing β-catenin accumulation, even at low concentrations. Our findings suggest that gallic acid holds substantial potential as a promising intervention to counteract metastatic features in breast cancer.

## Figures and Tables

**Figure 1 nutrients-15-03596-f001:**
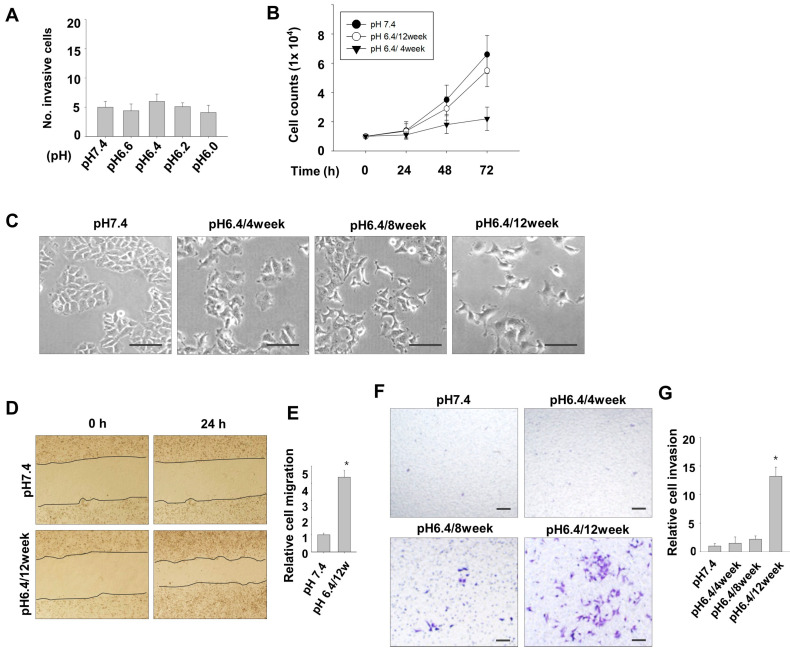
Long-term exposure to environmental acidity induces migration and invasive potential in MCF7 cells. (**A**) MCF7 cells exposed to a normal pH medium (pH 7.4) and various acidic pH-adjusted culture media (pH 6.6, 6.4, 6.2, and 6.0) were analyzed for invasive activity. The numbers of invaded cells are depicted graphically. (**B**) Cell growth was assessed by culturing MCF7 cells from the normal pH (7.4) condition, pH 6.4 for 4 weeks, and pH 6.4 for 12 weeks in a 24-well plate with regular medium for 3 d. (**C**) Phase-contrast images were captured for cells cultured under normal pH condition and those cultured at pH 6.4, for 4, 8, and 12 weeks. (**D**,**E**) Cells were cultured in either pH 7.4 or 6.4 medium for 12 weeks, and cell migration was evaluated using a wound healing assay at 24 h after creating the scratch (**D**), and the relative migration distance is graphically represented (**E**). (**F**,**G**) Cells cultured at pH 7.4 condition and those cultured at pH 6.4, for 4, 8, and 12 weeks were analyzed for invasion activity using a Matrigel-coated Transwell plate. Invaded cells were detected under a microscope (**F**), and the relative number of invaded cells are graphically represented (**G**). * *p* < 0.05 vs. pH 7.4. Scale bar = 100 μm.

**Figure 2 nutrients-15-03596-f002:**
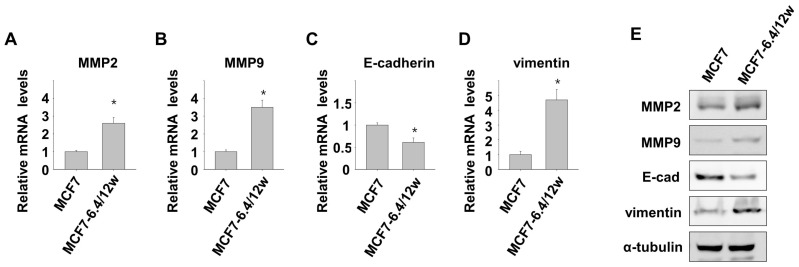
Long-term exposure to environmental acidity induces epithelial-to-mesenchymal transition (EMT)-like features. The mRNA expression of Matrix Metalloproteinases (MMP) 2, MMP9, E-cadherin, and vimentin was analyzed using real-time PCR (**A**–**D**) and Western blot assay (**E**) in MCF7 cells and MCF7 cells cultured for over 12 weeks at pH 6.4 (MCF7-6.4/12w cells). * *p* < 0.05 vs. pH 7.4.

**Figure 3 nutrients-15-03596-f003:**
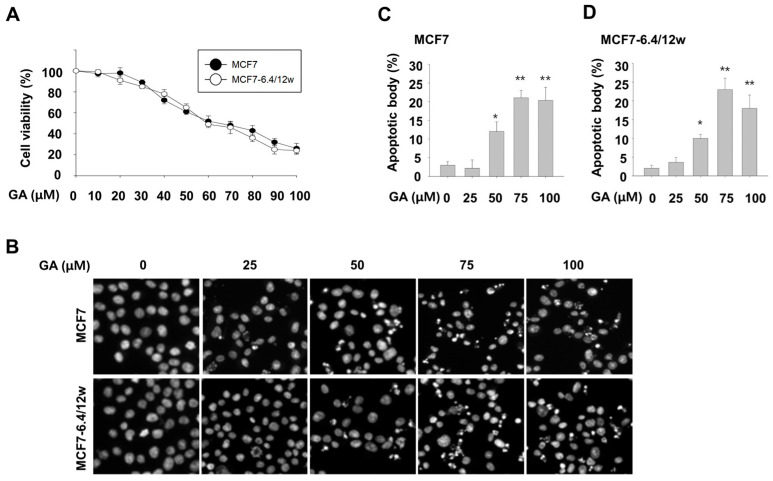
Gallic acid exerts inhibitory effect on the survival of both normal and acidity-adapted MCF7 cells. Cell viability was determined using the EZ-Cytox assay 48 h after treating MCF7 and MCF7-6.4/12w cells with gallic acid at specified concentrations (**A**). (**B**–**D**) Following treatment with gallic acid at different concentrations, MCF7 and MCF7-6.4/12w cells were stained with Hoechst 33342 to visualize apoptotic bodies under a fluorescent microscope (**B**), and the number of apoptotic nuclei in MCF7 (**C**) and MCF7-6.4/12w cells (**D**) was quantified. * *p* < 0.05, ** *p* < 0.01 vs. pH 7.4.

**Figure 4 nutrients-15-03596-f004:**
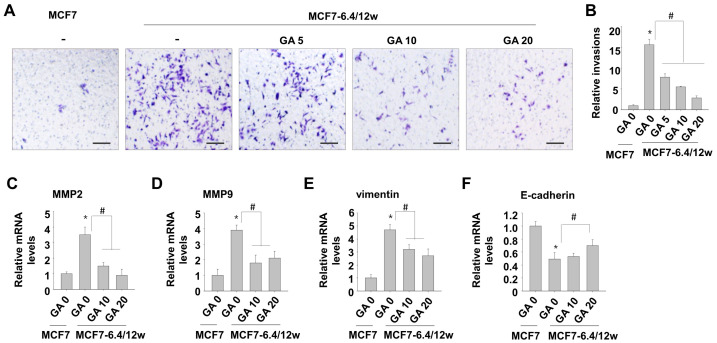
Low concentrations of gallic acid decreases acidity-induced metastatic characteristics in MCF7-6.4/12w cells. (**A**,**B**) Following 48 h of treatment with 5, 10, and 20 μM of gallic acid in MCF7-6.4/12w cells, an invasion assay was conducted (**A**) and the relative number of invaded cells compared to the normal MCF7 cells (**B**). (**C**–**F**) MCF7-6.4/12w cells were treated with 0, 10, and 20 μM of gallic acid for 48 h, and the mRNA expression levels of MMP2 (**C**), MMP9 (**D**), vimentin (**E**), and E-cadherin (**F**), and were analyzed using real-time PCR. The relative mRNA expression levels were calculated with the number of untreated normal MCF7 cells as the reference. * *p* < 0.05 vs. MCF7, # *p* < 0.05 vs. untreated MCF7-6.4/12w. Scale bar = 100 μm.

**Figure 5 nutrients-15-03596-f005:**
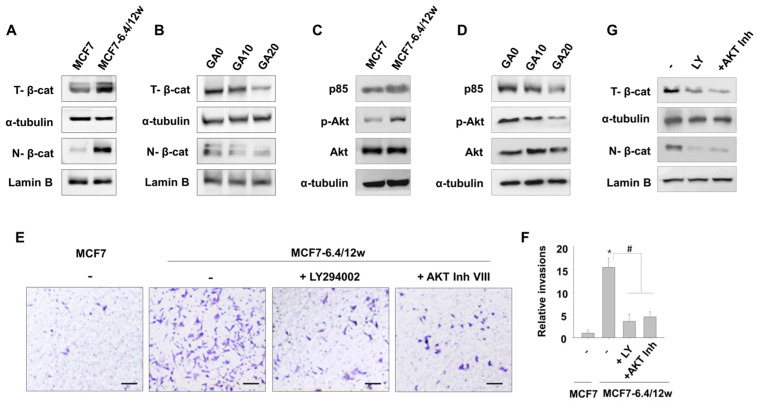
The inhibitory effect of gallic acid on the invasion of MCF7-6.4/12w cells is related to the inhibition of the PI3K/Akt signaling-controlled β-catenin pathway. (**A**–**D**) Western blot assay was used to analyze total and nuclear β-catenin levels in MCF7 and MCF7-6.4/12w cells (**A**) and to assess the expression levels of p85, pAKT, and total AKT (**C**). In MCF7-6.4/12w cells, after treatment with 0, 10, and 20 μM of gallic acid, total and nuclear β-catenin levels (**B**), along with the expression levels of p85, pAKT, and total AKT (**D**), were also evaluated using a Western blot assay. (**E**,**F**) An invasion assay was conducted using normal MCF7 cells and MCF7-6.4/12w cells treated with either 1 μM LY294002, 1 μM AKT Inhibitor VIII, or untreated (**E**), and the relative ratio of invaded cells compared to normal MCF7 cells is represented (**F**). (**G**) MCF7-6.4/12w cells treated with 1 μM LY294002, 1 μM AKT Inhibitor VIII, or untreated, were analyzed by Western blot assay to detect total and nuclear β-catenin levels. Lamin B is nuclear loading control. * *p* < 0.05 vs. MCF7, # *p* < 0.05 vs. untreated MCF7-6.4/12w. Scale bar = 100 μm.

**Figure 6 nutrients-15-03596-f006:**
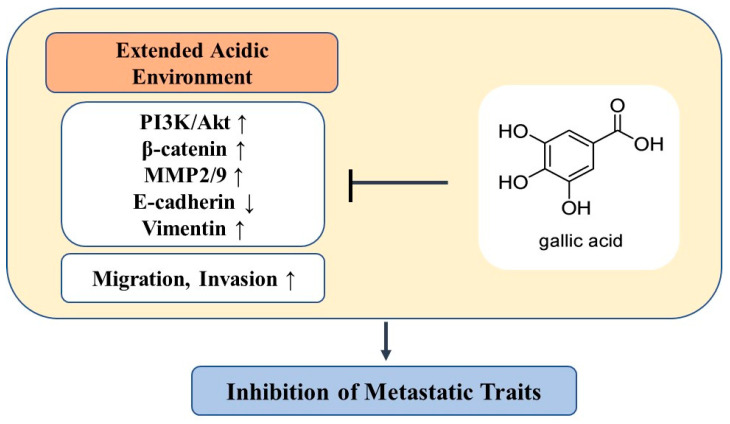
A schematic model illustrating the role of gallic acid in inhibiting metastatic traits induced by prolonged environmental acidity. ↑: upregulation, ↓: downregulation.

## Data Availability

Not applicable.

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
