# Peer review of "Anticancer Effect of Gallic Acid on Acidity-Induced Invasion of MCF7 Breast Cancer Cells"

_nutrients, 2023, doi:10.3390/nu15163596_

Round 1
Reviewer 1 Report
The manuscript "Anti-cancer Effect of Gallic Acid on Acidity-Induced Invasion of MCF7 Breast Cancer Cells" presents a relevant proposal by testing the use of gallic acid and its possible anti-cancer activity in breast cancer cells.
Below my considerations:
- Lines 67-69: "Additionally, we investigated the potential of gallic acid to counteract these metastatic traits and explored the underlying mechanisms" Explain what these mechanisms would be.
- Line 77: Explain why the pH of the cell culture was adjusted to this specific pH.
- In the results the pH variation cited in lines 159 and 160 was not mentioned in the methodology.
- The visualization of the figures is not clear, making it difficult to analyze the results. The size and resolution need to be improved to favor understanding. In addition, some images may need color for better visualization.
- The conclusions could be better elaborated by exploring the results found and addressing future perspectives.
Author Response
We appreciate for reviewing our manuscript again and giving us valuable information. Our manuscript has been revised as followings as indicated by Reviewers’ Comments. The modified sentences are highlighted in red in the manuscript.
Response to reviewers Comments:
The manuscript "Anti-cancer Effect of Gallic Acid on Acidity-Induced Invasion of MCF7 Breast Cancer Cells" presents a relevant proposal by testing the use of gallic acid and its possible anti-cancer activity in breast cancer cells.
Below my considerations:
- Lines 67-69: "Additionally, we investigated the potential of gallic acid to counteract these metastatic traits and explored the underlying mechanisms" Explain what these mechanisms would be. We added an explanation in the text, indicating that gallic acid holds the potential to alleviate metastatic traits through its capacity to suppress the underlying mechanisms—specifically, the elevated PI3K/Akt pathway and the nuclear accumulation of β-catenin. Lines 67-71.
- Line 77: Explain why the pH of the cell culture was adjusted to this specific pH. We sought to determine whether prolonged exposure to acidic culture conditions induces invasive capabilities in MCF7 cells. The cells were maintained under acidic pH conditions of 6.6, 6.4, 6.2, and 6.0 for several weeks, with subculturing every 4-5 days. Howeever, during this process, cells cultured under acidic conditions at pH 6.0 and 6.2 exhibited cell death after an extended period, making long-term exposure unfeasible at these pH levels. Thus, a prolonged study was conducted at pH 6.4, the lowest pH at which cell viability could be sustained. We included this explanation in the manuscript. Lines 166-172.
- In the results the pH variation cited in lines 159 and 160 was not mentioned in the methodology. We mentioned the preparation of various pH media in the ‘materials and methods’. Lines 78-79.
- The visualization of the figures is not clear, making it difficult to analyze the results. The size and resolution need to be improved to favor understanding. In addition, some images may need color for better visualization. We improved the size and resolution of figure images, and changed invasion images to color, as recommended.
- The conclusions could be better elaborated by exploring the results found and addressing future perspectives. In accordance with the recommendations, we enhanced the size and resolution of figure images and transformed the result images of invasion experiments into color images. For instances where it was possible to increase the original image resolution, we utilized the augmented images, while for cases where higher resolution was unavailable, we substituted them with alternate images saved at high resolutions.
- The conclusions could be better elaborated by exploring the results found and addressing future perspectives. We have revised the conclusion part, as recommended.
Reviewer 2 Report
The manuscript (nutrients-2567598) entitled with “Anti-cancer Effect of Gallic Acid on Acidity-Induced Invasion of MCF7 Breast Cancer Cells” by Ran Hong et al, was recently submitted to “Nutrients” as a research article for possible consideration. Overall, there are some typos and gramma errs. All Figures in the manuscript were rather too vague to tell.
Strongly suggest the authors try to replace those Figures with high-resolution files.
Moderate editing of English language required.
Typos and slight gramma errs should be avoided.
Author Response
We appreciate for reviewing our manuscript again and giving us valuable information. Our manuscript has been revised as followings as indicated by Reviewers’ Comments. The modified sentences are highlighted in red in the manuscript.
Response to reviewers Comments:
The manuscript (nutrients-2567598) entitled with “Anti-cancer Effect of Gallic Acid on Acidity-Induced Invasion of MCF7 Breast Cancer Cells” by Ran Hong et al, was recently submitted to “Nutrients” as a research article for possible consideration. Overall, there are some typos and gramma errs. All Figures in the manuscript were rather too vague to tell. We checked typos and errors again, and all figures are substituted with high-resolution images.
Strongly suggest the authors try to replace those Figures with high-resolution files.
We enhanced the size and resolution of figure images and transformed the result images of invasion experiments into color images. In cases where enhancing the original image resolution was feasible, we incorporated the augmented images. For situations where higher resolution was unavailable, we replaced them with alternate images saved at high resolutions.
We also have revised the manuscript to enhance the clarity of the introduction, results, and conclusion sections.
Reviewer 3 Report
This paper “Anti-cancer Effect of Gallic Acid on Acidity-Induced Invasion of MCF7 Breast Cancer Cells” aimed to determine whether non-metastatic MCF7 breast cancer cells can develop metastatic properties under acidic conditions. Additionally, the authors investigated the potential of gallic acid to counteract these metastatic traits and explored the underlying mechanisms. The article presents sufficient background investigation with reasonable data analysis, which are in line with the readers’ interest of Nutrients. However, there are still some shortcomings that need to be further improved or explained.
Comments:
Q1. The resolution of all graphics is insufficient and requires enhancement.
Q2. The clarity of the pictures hinders the analysis of pertinent expressions in the paper.
Q3. In Figure 5A, the two lanes are not in close proximity. Why is there an absence of duplicated data for modification?
Q4. The supplementation of the signal pathway diagram studied in this paper is recommended to enhance the elucidation of the research significance.
Author Response
We appreciate for reviewing our manuscript again and giving us valuable information. Our manuscript has been revised as followings as indicated by Reviewers’ Comments. The modified sentences are highlighted in red in the manuscript.
Response to reviewers Comments:
Q1: We enhanced the size and resolution of graphic images.
Q2: To improve the clarity of images, we enhanced the size and resolution of figure images and transformed the result images of invasion experiments into color images. In cases where enhancing the original image resolution was feasible, we incorporated the augmented images. For situations where higher resolution was unavailable, we replaced them with alternate images saved at high resolutions.
Q3: In fact, in fig5A, the removed lanes do not indicate duplicated data; rather, they represent different conditions (pH 6.6/12 weeks) that have been excluded. Since these conditions are not mentioned in this paper, they can be disregarded.
Q4: As suggested by a reviewer, we have added a schematic model illustrating the role of gallic acid in inhibiting metastatic traits as Figure 6.
Round 2
Reviewer 3 Report
The resolution of the picture in the paper has not been effectively enhanced. It is recommended to submit the original images to the editorial department for further improvement. I have no additional inquiries.